# StackMix: A complementary Mix algorithm

John Chen[1]    Samarth Sinha[2]    Anastasios Kyrillidis[1]

[1]Computer Science Dept., Rice University, USA
[2]Computer Science Dept., University of Toronto, Canada

## Abstract

Techniques combining multiple images as input/output have proven to be effective data augmentations for training convolutional neural networks. In this paper, we present StackMix: each input is presented as a concatenation of two images, and the label is the mean of the two one-hot labels. On its own, StackMix rivals other widely used methods in the "Mix" line of work. More importantly, unlike previous work, significant gains across a variety of benchmarks are achieved by combining StackMix with existing Mix augmentation, effectively mixing more than two images. E.g., by combining StackMix with CutMix, test error in the supervised setting is improved across a variety of settings over CutMix, including 0.8% on ImageNet, 3% on Tiny ImageNet, 2% on CIFAR-100, 0.5% on CIFAR-10, and 1.5% on STL-10. Similar results are achieved with Mixup. We further show that gains hold for robustness to common input corruptions and perturbations at varying severities with a 0.7% improvement on CIFAR-100-C, by combining StackMix with AugMix over AugMix. On its own, improvements with StackMix hold across different number of labeled samples on CIFAR-100, maintaining approximately a 2% gap in test accuracy –down to using only 5% of the whole dataset– and is effective in the semi-supervised setting with a 2% improvement with the standard benchmark Π-model. Finally, we perform an extensive ablation study to better understand the proposed methodology.

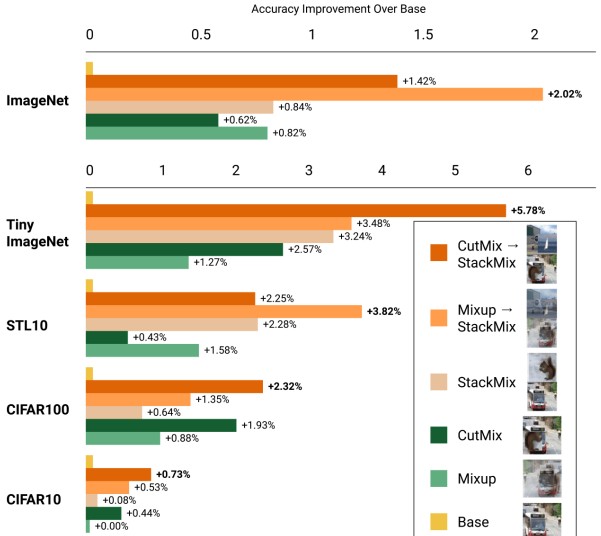

Figure 1: Base is the typical data augmentation setting, with random crops, horizontal flips, and normalization. Improvement over base refers to the attained test set accuracy minus the base test set accuracy. A→B refers to generating inputs with A and then feeding them as input to B. StackMix variants perform the best, and exhibit complementary behavior with existing augmentation.

## 1 INTRODUCTION

In the last decade, numerous innovations in deep learning for computer vision have substantially improved results on

*Accepted for the 38th Conference on Uncertainty in Artificial Intelligence* (UAI 2022).

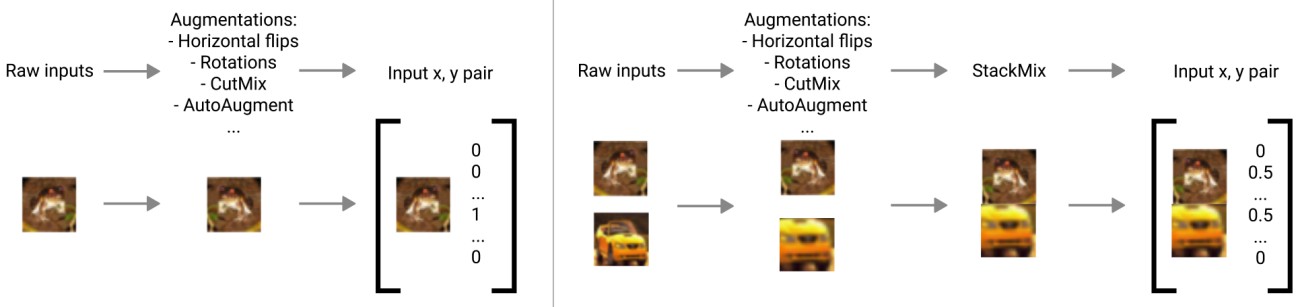

Figure 2: The StackMix procedure for $k = 2$. Left: The standard one-hot training. Right: StackMix with two images. Top row: Abstract pipeline. Bottom row: Concrete example.

many benchmark tasks [Krizhevsky et al., 2012, He et al., 2016, Zagoruyko and Komodakis, 2016, Huang et al., 2017]. These innovations include architecture changes, training procedure improvements, data augmentation techniques, and regularization strategies, among many others. In particular, data augmentation techniques have consistently and predictably improved neural network performance, and remain crucial in training deep neural networks effectively.

One such recent line of work revolves around the idea of finding effective augmentations through a search procedure [Cubuk et al., 2018]. The resulting augmentations tend to outperform hand-designed algorithms [Cubuk et al., 2018], and have seen some adoption [Tan and Le, 2020, Berthelot et al., 2019]. There is work to reduce the cost of the search [Lim et al., 2019, Ho et al., 2019].

A different line of work follows the idea of Mixup [Zhang et al., 2017], where inputs are generated from convex combinations of images and their labels. The resulting image can be understood as one image overlaid on another, with some opacity. Follow up works include methods such as Cutout [DeVries and Taylor, 2017] where parts of an image are removed, or CutMix [Yun et al., 2019] where parts of one image are removed and pasted onto another, with correspondingly weighted labels. Other works further improve accuracy [Kim et al., 2020], or robustness [Hendrycks et al., 2020]. While highly effective individually, these methods generally cannot be combined with each other (See MixUp→CutMix and CutMix→MixUp in Tables 3,4). Furthermore, many cannot effectively combine more than 3 images (see Table 10), or they suffer from information loss due to inappropriate occlusion.

In this paper, we consider the supervised setting and introduce StackMix, a complementary Mix algorithm. In StackMix, each input is presented as a concatenation of two images, and the label is the mean of the two one-hot labels. We show StackMix works well with existing tuned hyperparameters, and has no change to existing losses or general network architecture, which allows for easy adoption and integration into modern deep learning pipelines. Most importantly, not only is StackMix an effective augmentation on its own, it can further boost the performance of existing data augmentation, including the Mix line of work.

Our findings can be summarized as follows:

- Compared to the vanilla case, StackMix improves the test performance on existing image classification tasks, including by 0.84% on ImageNet with ResNet-50, 3.24% on Tiny ImageNet with ResNet-56 [He et al., 2016], 1.30% on CIFAR-100 with VGG-16 [Simonyan and Zisserman, 2014] and 0.64% with PreAct ResNet-18, 0.08% on CIFAR-10 with SeResNet-18 and 0.14% with ResNet-20, 2.28% on STL-10 with Wide-ResNet 16-8 [Zagoruyko and Komodakis, 2016]. Finally, StackMix improves by 2.16% on CIFAR-10, with all but 4000 labeled samples, when combined with the semi-supervised Π-model [Laine and Aila, 2017].

- We demonstrate that StackMix is complementary to existing data augmentation techniques, achieving over 0.8% improvement on ImageNet, 3% improvement on Tiny ImageNet, 2% test error improvement on CIFAR-100, 0.5% on CIFAR-10, and 1.5% on STL-10, by combining StackMix with state-of-the-art data augmentation method CutMix [Yun et al., 2019], as compared to CutMix alone. Similar gains are achieved with MixUp [Zhang et al., 2017] and AutoAugment [Cubuk et al., 2018]. In this way, many images are effectively combined.

- Improvements carry over to the robustness setting, with 1% test error improvement on CIFAR100-C [Hendrycks and Dietterich, 2019] and 0.2% on CIFAR10-C, by combining StackMix with state-of-the-art data augmentation method for robustness Augmix [Hendrycks et al., 2020] over AugMix.

Some of these results are summarized in Figure 2.

| Experiment short name | One Hot | StackMix | % Diff. |
|---|---|---|---|
| RN50-IMAGENET | $25,557,032$ | $25,557,032$ | 0% |
| RN56-TINYIMAGENET | $1,865,768$ | $2,070,568$ | 10.9% |
| VGG16-CIFAR100 | $15,038,116$ | $15,300,260$ | 1.7% |
| PRN18-CIFAR100 (-AA/-INF) | $11,222,244$ | $11,222,244$ | 0.0% |
| SRN18-CIFAR10 | $11,267,842$ | $11,267,842$ | 0.0% |
| RN20-CIFAR10 | $570,602$ | $573,162$ | 0.4% |
| WRN-STL10 | $11,002,330$ | $11,048,410$ | 0.4% |
| WRN-CIFAR10/100-C | $2,255,156$ | $2,267,956$ | 0.5% |
| WRN-CIFAR10-SSL | $1,467,610$ | $1,467,610$ | 0.0% |
| RN20-CIFAR10-N | $570,602$ | $573,162$ | 0.4% |
| VGG16-CIFAR100-N | $15,038,116$ | $15,300,260$ | 1.7% |

Table 1: Model Parameters for each experiment - $k = 2$. Details of the experiments are provided in the Results section.

---

**Algorithm 1** The StackMix algorithm. Produces one sample. For concatenating two images, we set $k = 2$. To recover the standard one-hot supervised training, we set $k = 1$.

**Inputs**: Samples $\{x_i, y_i\}_{i=0}^k$; $x_i$ are inputs and $y_i$ are one-hot labels; stochastic transformation $T$; number of images to concatenate $k$.
   1. Compute $x_i = T(x_i)$.
   2. Concatenate as $x = \texttt{concat}\left(\{x_i\}_{i=0}^k\right)$
   3. Compute prediction $y = \frac{1}{k}\left(\sum_{i=0}^k y_i\right)$
**return** $x, y$

---

# 2 THE STACKMIX DATA AUGMENTATION

In StackMix, we alter the input to the network to be a concatenation of $k$ images, and the output to be a $k$-hot vector with values $1/k$ in active classes; see Figure 2 for the case of $k = 2$. *For clarity and simplicity, we will assume $k = 2$ for the rest of the paper.* The choice of $1/2$ value (for $k = 2$) is a result of using the Cross Entropy loss, and values 1 and 1 can be explored for the Binary Cross Entropy loss. StackMix is tightly related to the line of "Mix" data augmentation work. StackMix has the following general advantages:

(a) StackMix is complementary to existing data augmentations including the "Mix" line of work (see Section 3.4). StackMix can effectively mix more than two images, e.g. StackMix with $k = 2$ and Mixup can effectively mix four images in total. This is in contrast to, for example, Mixup which does not benefit from mixing more than two images (see Page 3 of Mixup [Zhang et al., 2017] or Table 10). In addition, the various "Mix" based methods cannot be effectively combined (See MixUp→CutMix, CutMix→MixUp in Tables 3,4).

(b) StackMix has no additional hyperparameters in terms of fine-tuning during training.

(c) Compared with methods which remove or replace parts of images, StackMix has no assumption that critical information is effectively captured in bounding boxes, which may not be the case for real-world datasets.

This construction can be directly plugged into any existing image classification training pipeline, with the only typical changes being the sizes of the first layer of the network. The change in parameters is generally insignificant (e.g., $< 1\%$ for ResNet-20 on CIFAR10, or 0% for PreAct ResNet-18 on CIFAR100, due to average pool; see Table 1; see Tables 3,4 for controls). To ensure fairness in comparisons, we tune hyperparameters in the original standard one-hot supervised setting –including epochs to ensure performance has saturated– and we then apply the **exact same hyperparameters** to StackMix. We note that, for testing, we concatenate

the same image twice, with the one-hot vector used as the ground truth label (see later section for discussion).

## 2.1 IMPLEMENTATION AND SYNERGY WITH EXISTING DATA AUGMENTATION

In the traditional setting, a batch size of $M$ is defined by having $M$ inputs per batch, where each of the $M$ inputs is typically the result after data augmentation. For consistency with data augmentation techniques, which combine two or more images such as Mixup [Zhang et al., 2017], we define an input vector as a vector after the concatenation. In particular, and for simplicity of presentation, for each input, we assume we perform the following motions (for $k = 2$):

(a) Sample two images.

(b) Apply existing data augmentation to each image individually.

(c) Concatenate the two images as a single input vector.

(d) Rescale each label vector to $1/2$, and add them element-wise to produce the multi-hot label.

This method can be easily extended to $k$-fold concatenation of images, where each label vector is rescaled to $1/k$, and then summed element-wise. We explore $k > 2$ in Section 3.5. For clarity, we present this procedure as well in Algorithm 1, where $k = 1$ is the standard one-hot training procedure, and $k = 2$ is the primary focus of this paper. In implementation, we sample two images with replacement and thus the output can be a one-hot vector, naturally with $1/n$ probability, where $n$ is the size of the dataset; although, we note that this choice has minimal impact on performance.

# 3 RESULTS

We provide results for $i$) supervised image classification, $ii$)test error robustness against image corruptions and perturbations, $iii$) semi-supervised learning, combining StackMix with existing data augmentation, $iv$) an ablation study, and $v$) evaluation of test time augmentation. A summary

of experimental settings are given in Table 2, and comprehensively detailed in each section. *We tuned the hyperparameters of the standard one-hot setting to achieve the performance of the original papers and of the most popular public implementations, reusing the most widely used codebases for consistency.* We then used the *exact same hyperparameters and pipeline* for StackMix for fairness.

## 3.1 SUPERVISED IMAGE CLASSIFICATION

In this section, we explore improving the performance of well-known baselines in the supervised learning setting. We add StackMix to seven model-dataset pairs, and lastly observe the performance with and without StackMix across a varying number of supervised samples in the CIFAR100 setting. See Tables 3,4 for results.

**Controls.** Although StackMix generally introduces a small number of additional parameters (see Table 1), it is crucial to introduce controls to account for the difference, in addition to the increased training time. Therefore, we also present results with two controls. To account for the additional hyperparameters and computation, we introduce a control where the StackMix procedure concatenates the same image with itself during training, after being individually augmented with the stochastic image augmentation for fairness. To account for increased training time, we introduce a control with double the batch size and double the epochs, with re-tuned learning rate. This way we effectively control for both the model size and the total computation/number of images seen by the model during training. Results are presented in Tables 3,4 (See Base, 2x bs/epochs, StackMix(same)). It appears that neither control exhibits the same improvement as with StackMix. *This suggests (but does not guarantee) the effect of StackMix is nontrivial and potentially cannot be explained by computation or model size differences.*

**Examining learned embeddings.** We check the learned embeddings for randomly drawn samples from CIFAR100 with t-SNE [Van Der Maaten and Hinton, 2008], given in Figure 3, as sanity check. The images are processed in the inference setting, where they are concatenated with themselves. Clusters form as expected (Figure 3 Left). This is highly encouraging despite the network mostly seeing the concatenation of images from different classes. We also observe that by fixing one image of each concatenation to be a certain class and varying the class of the other image, a similarly separated distribution forms (Figure 3 Middle). This further supports the idea that the network has learned to differentiate between the two presented images. Finally, we find that concatenating images from two different classes is semantically separated from concatenating either image with itself (Figure 3 Right), and that as a sanity check the embeddings are generally not sensitive to which image is placed on top. In sum, while the network sees the same image stacked during testing and largely sees different images stacked during training, it appears to learn reasonable embeddings.

**Results on ImageNet [Russakovsky et al., 2015].** We experiment with ResNet-50 [He et al., 2016]; we use the official PyTorch implementation and train the network for the default 90 epochs, which roughly follows popular works [He et al., 2016, Huang et al., 2017, Han et al., 2017, Simonyan and Zisserman, 2014, Zhang et al., 2017]. There are some works which train the network for 3-4x the number of epochs, e.g. CutMix [Yun et al., 2019], but this can be computationally demanding. We use the standard random crops of size $224 \times 224$, horizontal flips, and normalization. In inference, we use a $224 \times 224$ center crop, following standard. The network is trained with momentum SGD ($\eta = 0.1$, $\beta = 0.9$), with a 30-60 decay schedule by factor of 0.1 using a batch size of 256. We set $\alpha = 1$ for MixUp and CutMix. StackMix variants perform the best, with best variant improving 2.02% over base. Adding StackMix to MixUp improves 1.20% over MixUp, and adding StackMix to CutMix improves 0.80% over CutMix (Table 3 and Figure 1).

**Results on Tiny ImageNet.** With ResNet-56 [He et al., 2016], we trained the model for 80 epochs with momentum SGD ($\eta = 0.1$, $\beta = 0.9$), Cross Entropy loss, decaying by a factor of 0.1 at 40 and 60 epochs, using a batch size of 64. We applied the standard image augmentation [He et al., 2016] of horizontal flips, normalization and random crops. By adding StackMix to the vanilla case, the absolute generalization error was reduced by 3.24%, from 42.03% to 38.79%. By observing Figure 4 (Left), we see that while the two methods are initially comparable, adding StackMix reduces the error in the later stages of training. The plateau of the StackMix curve suggests resistance to overfitting. Furthermore, by adding StackMix to MixUp, test error is decreased by 2%, and by adding StackMix to CutMix, test error is decreased by 3% (Table 4 and Figure 1).

**Results on CIFAR100.** We trained two models, VGG16 and PreActResNet-18 (PRN18). VGG-16 was trained for 300 epochs following standard procedure as in Tiny ImageNet. PRN18 was trained similarly, except for 200 epochs and a learning rate decay schedule by a factor 0.2 at 60, 120, and 180 epochs. A roughly 1% test error improvement is observed for both cases for StackMix compared to the controls. Relative to MixUp and CutMix, a significant decrease of 1-2% is observed by adding StackMix (Table 4 and Figure 1). We observe in Figure 4 (Right) that StackMix already improves in the early stages of training with VGG16. It is typical in neural network training to see the gap closed in the first learning rate decay when there exists a gap early on in training, but here StackMix maintains an improvement.

**Results on CIFAR10.** We trained ResNet20 (RN20) and SeResNet-18 (SRN18). This is a particularly challenging task to improve upon due to the model architecture where doubling the number the parameters and increasing the

| Experiment short name | Model | Dataset | Setting |
|---|---|---|---|
| RN50-IMAGENET | ResNet-50 | ImageNet | Supervised Learning |
| RN56-TINYIMAGENET | ResNet-56 | Tiny ImageNet | Supervised Learning |
| VGG16-CIFAR100 | VGG-16 | CIFAR100 | Supervised Learning |
| PRN18-CIFAR100 | PreActResNet-18 | CIFAR100 | Supervised Learning |
| SRN18-CIFAR10 | SeResNet-18 | CIFAR10 | Supervised Learning |
| RN20-CIFAR10 | ResNet-20 | CIFAR10 | Supervised Learning |
| WRN-STL10 | Wide ResNet 16-8 | STL10 | Supervised Learning |
| WRN-CIFAR10-SSL | Wide ResNet 28-2 | CIFAR10 | Semi-Supervised Learning |
| WRN-CIFAR10/100-C | Wide ResNet 40-2 | CIFAR10/100-C | Robustness |
| RN20-CIFAR10-N | ResNet-20 | CIFAR10 | Ablation |
| VGG16-CIFAR100-N | VGG-16 | CIFAR100 | Ablation |
| PRN18-CIFAR100-INF | PreActResNet-18 | CIFAR100 | Test time inference augmentation |

Table 2: Summary of experimental settings.

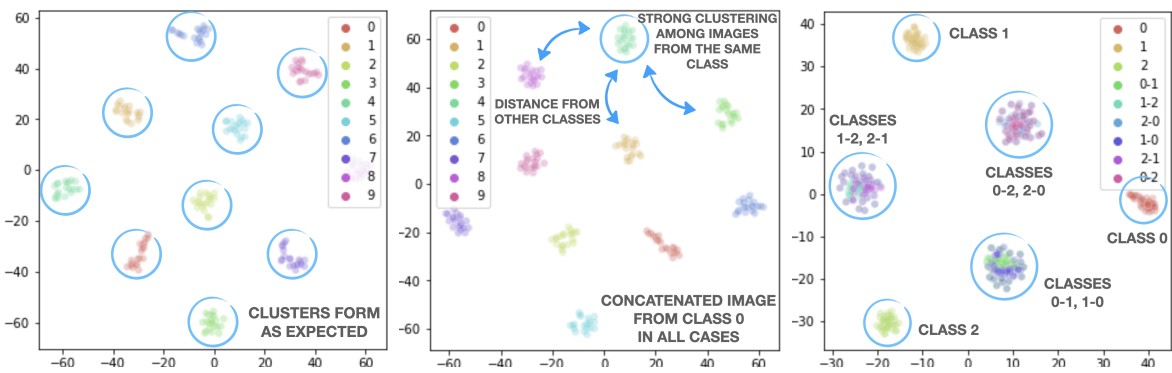

Figure 3: Left: Randomly selected 10 classes where each image is concatenated with itself. Middle: Randomly selected 10 classes where each image is concatenated with an image from class 0. Right: Randomly selected 3 classes where each image is concatenated with itself, concatenated as the top image with an image from another class, and concatenated as the bottom image with an image from another class. Singular number denotes self-concatenation. "a-b" denotes image from class a concatenated as the top image with an image from class b. Data used is training data and test data plots are similar with more noise.

| Method | Test error |
|---|---|
| Base | 24.10 |
| 2x bs/epochs | 24.49 |
| StackMix(same) | 23.28 |
| MixUp | 23.28 |
| CutMix | 23.48 |
| StackMix | 23.26 |
| MixUp→CutMix | 35.70 |
| CutMix→MixUp | 33.99 |
| MixUp→StackMix | **22.08** |
| CutMix→StackMix | 22.68 |

Table 3: Generalization error of experiments with and without StackMix in the RN50-ImageNet setting. 2x bs/epochs refers to doubling the batch size and epochs of Base. StackMix(same) is another control which refers to stacking the same image as input. A→B refers to generating inputs with A and then feeding them as input to B.

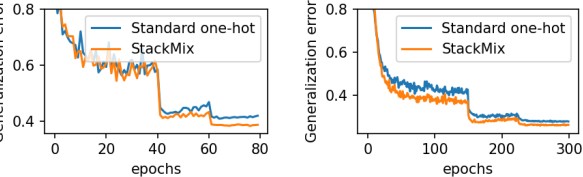

Figure 4: Generalization error for supervised learning. Left: RN56-TINYIMAGENET. Right: VGG16-CIFAR100.

depth results in only minor gains in performance [He et al., 2016]. Both networks are trained similarly as previously, except with a 30-60-90 learning rate decay schedule for SRN18. In both cases there are small improvements over the controls, and adding StackMix to existing augmentation further improves results. We emphasize that ResNet20 is not ResNet18, which is a different network architecture with significantly more parameters. Most popular implementations

| Experiment | Base | 2x bs/epochs | StackMix(same) | MixUp | CutMix | StackMix | MixUp→CutMix | CutMix→MixUp | MixUp→StackMix | CutMix→StackMix |
|---|---|---|---|---|---|---|---|---|---|---|
| RN56-TINYIMAGENET | 42.03 | 42.00 | 42.11 | 40.76 | 39.46 | 38.79 | 43.80 | 43.24 | 38.55 | **36.25** |
| VGG16-CIFAR100 | 27.80 | 28.63 | 27.69 | 27.35 | 27.20 | 26.50 | 34.44 | 35.66 | 25.69 | **25.49** |
| PRN18-CIFAR100 | 25.93 | 25.41 | 25.63 | 25.05 | 24.00 | 25.29 | 31.36 | 30.20 | 24.58 | **23.61** |
| RN20-CIFAR10 | 7.65 | 7.55 | 7.73 | 6.51 | 6.74 | 7.51 | 6.93 | 7.05 | 6.40 | **6.27** |
| SRN18-CIFAR10 | 5.03 | 5.05 | 5.21 | 5.34 | 4.59 | 4.95 | 6.64 | 6.59 | 4.50 | **4.30** |
| WRN-STL10 | 17.26 | 15.83 | 18.92 | 15.68 | 16.83 | 14.98 | 24.02 | 23.68 | **13.44** | 15.01 |

Table 4: Generalization error of experiments with and without StackMix in the supervised setting. 2x bs/epochs refers to doubling the batch size and epochs of Base. StackMix(same) is another control which refers to stacking the same image as input. A→B refers to generating inputs with A and then feeding them as input to B, e.g. CutMix→MixUp first generates inputs with CutMix and then feeds them as input to MixUp.

| samples% | 100 | 50 | 30 | 20 | 10 | 5 |
|---|---|---|---|---|---|---|
| Base | 27.80% ±.10 | 34.88% ±.20 | 42.52% ±.34 | 50.41% ±.38 | 71.91% ±.57 | 86.03% ±.12 |
| StackMix | **26.50% ±.11** | **33.61% ±.21** | **40.40% ±.34** | **48.19% ±.52** | **68.71% ±.87** | **85.61% ±.40** |

Table 5: Generalization error (%) for VGG16-CIFAR100 with varying number of proportional samples in each class.

of ResNet20 fall in the 8-8.5% test error range on CIFAR10. SeResNet18 is a variation of ResNet18. See Table 4.

**Results on STL10.** We use Wide ResNet 16-8 [Zagoruyko and Komodakis, 2016], a 16 layer deep ResNet architecture with 8 times the width. We trained the WRN model for 100 epochs following standard settings as before. Similar results are observed in Table 4, except CutMix does not perform as well, leading to MixUp→StackMix attaining the best performance.

**Understanding performance with varying labeled samples.** StackMix performs well in the above supervised settings, and we further explore performance in the low sample regime. In particular, we select the VGG16-CIFAR100 setting, and decrease the number of samples in each class proportionally. We use the exact same training setup as in the full VGG16-CIFAR100 case, and tabulate results in Table 5. We perform 3 runs since low-sample settings produce higher variance results. We see that similar improvements hold with lower samples at roughly 2% generalization error.

### 3.2 ROBUSTNESS

**Results on CIFAR10/100-C.** We investigate the impact of StackMix on robustness. We select a corrupted dataset as test set and reevaluate models trained with and without StackMix on the uncorrupted training set, following standardized procedure [Hendrycks and Dietterich, 2019]. AugMix [Hendrycks et al., 2020] follows the Mix line of work with significant increases in robustness, and we consider AugMix as the state-of-the-art baseline. Results with WRN-40-2 on CIFAR100-C and CIFAR10-C are shown in Tables 6 and 7.

In the clean case, StackMix improves over the vanilla case, and adding StackMix to AugMix significantly decreases test error, including 2% on CIFAR100. StackMix does not ap-

pear to provide any additional robustness improvements beyond improvements carried over from the clean case. However, this should not be taken for granted as some methods which increase robustness can lower clean test error and vice versa [Raghunathan et al., 2019]. AugMix→StackMix outperforms AugMix in both clean and corrupted cases on average, and outperforms AugMix in 11/15 categories in CIFAR10-C and 10/15 categories in CIFAR100-C.

### 3.3 SEMI-SUPERVISED LEARNING

We explore if StackMix can be directly applied to improve Semi-Supervised Learning (SSL) [Chapelle and Scholkopf, 2006], where the network processes both labeled and unlabeled samples. We select a popular and practical subset of SSL, which involves adding a loss function for consistency regularization [Tarvainen and Valpola, 2017, Berthelot et al., 2019, Laine and Aila, 2017, Chen et al., 2020]. Consistency regularization is similar to contrastive learning in that it tries to minimize the difference in output between similar samples. In particular, we select the classic and standard benchmark of the Π-model [Laine and Aila, 2017].

The Π model adds a loss function for the unlabeled samples of the form: $d(f_\theta(x), f_\theta(\hat{x}))$, where $d$ is typically the Mean Square Error, $f_\theta$ is the output of the neural network, and $\hat{x}$ is a stochastic perturbation of $x$. Minimizing this loss enforces similar output distributions of an image and its perturbation. A coefficient is then applied to the SSL loss as a weight with respect to the Cross Entropy loss. The unlabeled samples are evaluated with the SSL loss, while the labeled samples are evaluated with Cross Entropy.

**Results on CIFAR10.** We follow the standard setup in [Oliver et al., 2018] for the CIFAR10 dataset, where 4000 labeled samples are selected, and remaining samples are unlabeled. We use a WRN 28-2 architecture [Zagoruyko

| | | Noise | | | Blur | | | | Weather | | | | Digital | | | | |
|---|---|---|---|---|---|---|---|---|---|---|---|---|---|---|---|---|---|
| Setting | Clean | Gauss. | Shot | Impulse | Defocus | Glass | Motion | Zoom | Snow | Frost | Fog | Bright | Contrast | Elastic | Pixel | JPEG | mCE |
| Standard | 25.18 | 83.1 | 75.2 | 75.8 | 43.2 | 78.4 | 49.6 | 50.4 | 48.3 | 53.5 | 39.6 | 30.0 | 48.1 | 44.1 | 54.4 | 55.0 | 55.24 |
| StackMix | 24.80 | 81.4 | 73.8 | 74.0 | 44.0 | 77.5 | 49.8 | 51.3 | 46.7 | 52.8 | 37.6 | 29.6 | 47.7 | 44.1 | 55.7 | 55.7 | 54.78 |
| AugMix | 23.37 | **54.5** | **46.7** | 36.7 | 25.6 | **53.1** | 29.6 | 28.2 | 34.7 | 36.0 | 33.0 | 25.6 | 31.3 | 32.4 | **36.5** | **38.3** | 36.14 |
| AugMix→StackMix | **21.81** | 55.7 | 47.3 | **33.0** | **24.0** | 54.9 | **27.8** | **27.2** | **32.8** | **34.9** | **29.9** | **23.9** | **30.9** | **31.3** | 39.3 | 39.0 | **35.46** |

Table 6: Generalization error of experiments with and without StackMix in the `WRN-CIFAR100-C` setting.

| | | Noise | | | Blur | | | | Weather | | | | Digital | | | | |
|---|---|---|---|---|---|---|---|---|---|---|---|---|---|---|---|---|---|
| Setting | Clean | Gauss. | Shot | Impulse | Defocus | Glass | Motion | Zoom | Snow | Frost | Fog | Bright | Contrast | Elastic | Pixel | JPEG | mCE |
| Standard | 5.49 | 51.1 | 39.1 | 43.1 | 19.1 | 50.5 | 24.2 | 25.2 | 19.1 | 23.2 | 11.9 | 7.1 | 21.8 | 16.7 | 30.0 | 22.6 | 26.98 |
| StackMix | 5.30 | 53.0 | 42.5 | 43.1 | 18.2 | 49.4 | 23.6 | 23.8 | 18.5 | 21.7 | 11.7 | 6.8 | 22.3 | 17.5 | 29.7 | 23.0 | 26.99 |
| AugMix | 4.91 | **22.2** | **16.3** | 13.0 | 5.8 | **21.0** | 7.9 | 7.2 | 10.7 | 10.4 | 8.2 | 5.6 | 7.8 | 10.1 | 16.3 | 12.6 | 11.67 |
| AugMix→StackMix | **4.37** | 24.2 | 16.8 | **10.7** | **5.3** | 22.4 | **7.5** | **6.7** | **9.9** | **10.0** | **7.5** | **5.0** | **7.3** | **9.6** | **16.2** | **12.5** | **11.44** |

Table 7: Generalization error of experiments with and without StackMix in the `WRN-CIFAR10-C` setting.

| Experiment | Base | StackMix |
|---|---|---|
| `WRN-CIFAR10-SSL` | 17.31% | **15.15%** |

Table 8: Generalization error of Π-model on the standard benchmark of CIFAR10, with all but 4,000 labels removed.

| Base | AA | StackMix | AA→StackMix |
|---|---|---|---|
| 25.93% | 23.87% | 25.29% | **21.51%** |

Table 9: Generalization error (%) of `PRN18-CIFAR100` with AutoAugment. It is no surprise that StackMix is complementary to AutoAugment, and we simply present one experiment here to confirm.

and Komodakis, 2016], training for 200,000 iterations with a batch size of 200, of which 100 are labeled and 100 are unlabeled. The Adam optimizer is used ($\eta = 3 \cdot 10^{-4}, \beta_1 = 0.9, \beta_2 = 0.999$), decaying learning rate schedule by a factor of 0.2 at 130,000 iterations. Horizontal flips, random crops, and gaussian noise are used as data augmentation. A coefficient of 20 is used for the SSL loss. By adding StackMix, we reduce the test error by 2.16% (see Table 8).

### 3.4 STACKMIX IS COMPLEMENTARY TO EXISTING DATA AUGMENTATION

Results in the previous section strongly suggest that Stack-Mix is complementary to Mix methods, with improved training by combining StackMix with MixUp [Zhang et al., 2017], CutMix [Yun et al., 2019] and AugMix [Hendrycks et al., 2020]. This differs from existing work: e.g., MixUp and CutMix cannot be effectively combined (see Tables 3,4). We further support the complementary nature of StackMix by combining with AutoAugment [Cubuk et al., 2018].

**Results on CIFAR100.** We follow the settings in `PRN18-CIFAR100`. We use existing AutoAugment poli-

cies for the CIFAR datasets, and following [Cubuk et al., 2018] for CIFAR, we apply AutoAugment after other augmentations, and before normalization and StackMix. AutoAugment improves 2% over standard augmentation, and adding StackMix improves by another 2% (see Table 9); again, suggesting a complementary behavior and easy incorporation into existing pipelines. We want to emphasize the result in this section. *A 2% gain by combining StackMix with AutoAugment over either baseline on the CIFAR100 dataset is comparable to a significant increase in model size and depth; on CIFAR100, typically moving from a ResNet18 model to ResNet101 and beyond on yields a roughly 2% improvement in most implementations.*

### 3.5 ABLATION STUDY

We now perform an ablation study to determine how far this framework can be pushed. We increase the value of $k$, and observe the test error in the setting of `VGG16-CIFAR100` and `RN20-CIFAR100`. We fix the hyperparameters as used previously, with results in Table 10 and Figure 5. For MixUp and CutMix, $k$ represents the number of images combined. For MixUp/CutMix→StackMix, $k$ represents the number of images stacked, after they have been pairwise augmented with MixUp/CutMix (e.g. $k = 3$ would be $2 \times 3 = 6$ total). We reduce the box size of CutMix to allow for higher $k$.

In almost all cases, the error deteriorates immediately after $k = 2$, and further increasing $k$ typically increases the error further, clearer in the case of `VGG16-CIFAR100`. Performance deterioration is significantly more severe for MixUp and CutMix, whereas the StackMix variants suffer only slightly. This is likely due to loss of semantic information with inputs looking similar for MixUp, and a failure to capture enough critical information for CutMix. For example, on CIFAR100 MixUp and CutMix almost double in error from $k = 2$ to 5, tenfold the error rate increase of the StackMix variants. We can see in Figure 5 that the

| Dataset-Model | Augmentation | 1 (base) | 2 (same image) | $k$ | | |
|---|---|---|---|---|---|---|
| | | | | 2 | 3 | 5 |
| VGG16-CIFAR100 | Standard | 27.80% | 27.69% | **26.50%** | 27.35% | 29.35% |
| | MixUp | - | - | **27.35%** | 49.04% | 63.52% |
| | CutMix | - | - | **27.20%** | 36.52% | 51.25% |
| | MixUp→StackMix | - | - | **25.69%** | 26.05% | 27.41% |
| | CutMix→StackMix | - | - | **25.49%** | 26.95% | 27.91% |
| RN-CIFAR10 | StackMix | 7.65% | 7.73% | **7.51%** | 8.13% | 7.89% |
| | MixUp | - | - | **6.51%** | 8.93% | 13.86% |
| | CutMix | - | - | **6.74%** | 7.63% | 9.91% |
| | MixUp→StackMix | - | - | 6.40% | 6.30% | **6.05%** |
| | CutMix→StackMix | - | - | **6.27%** | 6.65% | 6.81% |

Table 10: Generalization error for VGG16-CIFAR100 and RN20-CIFAR10 with varying number of images concatenated.

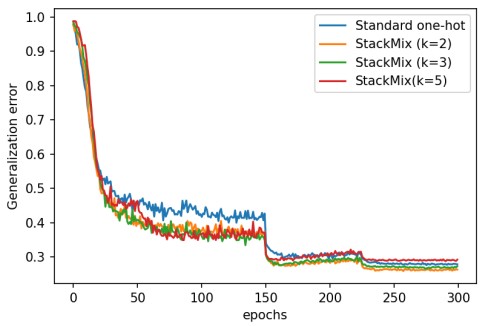 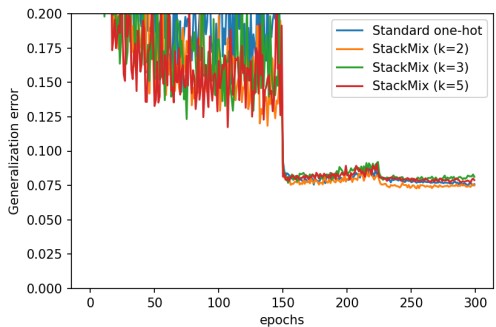

Figure 5: Left: Generalization error for VGG16-CIFAR100. Right: Generalization error for RN20-CIFAR10. Varying number of images concatenated.

| Base | Base + flips | StackMix | StackMix + flips |
|---|---|---|---|
| 25.93% | 25.36% | 25.29% | 24.79% |

Table 11: PRN18-CIFAR100-INF - Generalization error with test time augmentation. Base + flips is the mean output of an image and its flipped version. StackMix + flips is the output of the concatenation of an image with itself flipped.

choice of $k$ has limited impact in the early stages of training, but affects the final test error, where performance begins to deteriorate after the first learning rate decay.

Furthermore, we highlight results on the concatenation of the same image (also in Tables 3,4). First, this results in a sanity check that the StackMix construction on the same image is identical (with respect to performance) to the one-hot vector classification constructions. Second, worse performance in StackMix when the same image is concatenated twice indicates that the network learns less, as compared to the concatenation of two images: this further strengthens the effect that StackMix brings during training.

## 3.6 INFERENCE SPEED AND AUGMENTATIONS

One drawback of StackMix compared with the standard one-hot is slower inference speed due to the larger input size. Therefore, we designed several experiments.

On SRN18-CIFAR10, we only swept over the top image in StackMix for the first convolutional layer. This led to similar results to StackMix, with only 0.01 error difference.

In another case, the standard one-hot setup is given two forward passes for inference at test time. Concretely, we take the top-1 of the mean output of an image and its flipped counterpart. For StackMix in this paper, we concatenated the same image with itself without any further augmentation. However, we observe that the benefits of test-time augmentation for the standard case carry over to StackMix naturally without additional computation, where an image is concatenated with a itself flipped. The improvements with respect to each vanilla case are similar, where the standard case gains 0.57% and StackMix gains 0.50% (See Table 11).

Finally, we upsampled the base case images to account for the additional computation from the larger image in

StackMix. On `PRN18-CIFAR100`, a corresponding larger image size (45x45≈32x32x2) worsened error from the base case, from 25.93 to 27.15, whereas StackMix achieves 25.29 error. On `SRN18-CIFAR10`, using the corresponding larger image size achieved similar error on the base case, from 5.03 to 5.05, whereas StackMix achieves 4.95 error. These results suggest that the extra compute from larger input sizes is not the reason StackMix achieves gains.

## 4 RELATED WORK

In supervised learning, techniques that can be added to the label, such as label smoothing [Sukhbaatar et al., 2014], or directly to the data, using data augmentation [Zhang et al., 2016, Cubuk et al., 2018, DeVries and Taylor, 2017, Yun et al., 2019], or both [Zhang et al., 2017], boost performance. Horizontal image flips and crops are well-established as effective data augmentation techniques [Krizhevsky et al., 2012, He et al., 2016]. The choice of single-image augmentations was discovered through a search procedure [Cubuk et al., 2018]. The cost of the method was reduced further in [Ho et al., 2019, Lim et al., 2019].

StackMix is tightly related to the "Mix" line of work [Zhang et al., 2017, Yun et al., 2019, DeVries and Taylor, 2017, Hendrycks et al., 2020, Kim et al., 2020, Guo et al., 2018], where pairs of input images and their labels are combined. Mixup [Zhang et al., 2017] takes convex combinations of inputs and their labels, and has been extended to the feature space [Verma et al., 2019]. Other work removes parts of images [DeVries and Taylor, 2017, Zhong et al., 2017], and paste parts of images with weighted labels [Yun et al., 2019, Takahashi et al., 2020]. PuzzleMix [Kim et al., 2020] improves the salient information in Mix images, while AugMix [Hendrycks et al., 2020] improves robustness.

Ensembles [Dietterich, 2000] and multiple choice learning [Guzman-Rivera et al., 2012] output multiple labels from a single image. Ensembles utilize multiple models, while multiple choice learning predicts multiple labels. StackMix is strictly different from both as our aim is to predict multiple outputs from multiple inputs.

## 5 CONCLUSION

We introduce StackMix, a complementary Mix algorithm. StackMix can directly be plugged into existing pipelines with minimal changes: no change in loss, hyperparameters, or general network architecture. StackMix improves performance in a variety of benchmarks. StackMix is complementary to and boosts the performance of existing augmentation, including MixUp, CutMix, AugMix, and AutoAugment.

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
