# OpenReview forum: "StackMix: A complementary Mix algorithm"
_auai.org/UAI/2022/Conference — UAI 2022 Poster_

### Official Review · Reviewer_ibgi · 2022-04-11

**Q2(1) Originality/Novelty:** 2
**Q2(2) Significance/Impact:** 3
**Q2(3) Correctness/Technical Quality:** 3
**Q2(6) Clarity Of Writing:** 3
**Q6 Overall Score:** 6
**Q8 Confidence In Your Score:** 4

**Q1 Summary And Contributions:**

A new data augmentation method, StackMix, is proposed. StackMix works by concatenating two images (or more) and the mean of their corresponding one-hot label vector. The proposed is complementary to other augmentation methods, i.e., it can be used on top of other methods. The methods show performance gain in a wide range of experiments.


**Q2 Assessment Of The Paper:**

More detailed information regarding each of these aspects is given below:

**Q2(4) Quality Of Experiments (Optional):**

4: Excellent: The experimental evaluation is comprehensive and the results are compelling.

**Q2(5) Reproducibility:**

2: Fair: Key resources (e.g., proofs, code, data) are unavailable but key details (e.g., proof sketches, experimental setup) are sufficiently well-described for an expert to confidently reproduce the main results.

**Q3 Main Strengths:**

1. The method is simple and easy to implement.

2. It can be used with other existing augmentation methods.

3. The authors validated the method on multiple datasets, showing performance gain.

4. To provide a fair comparison, the authors accounted for a different number of hyperparameters, epochs, etc., showing that the effect of StackMix is nontrivial and potentially can not be explained by computation or model size differences.


**Q4 Main Weakness:**

Weakness and suggestions:

1. “In Stack-Mix, each input is presented as a concatenation of two images, and the label is the mean of the two one-hot labels.” Although the figure explains the axis of concatenation, it will be better if you mention the axis of concatenation in the text as well.

2. You should evaluate performance for unsupervised/SSL methods, which rely heavily on data augmentation techniques.

3. Section 3.6 is not very clear. Some of the experiments are not well explained, and the intuition is not clear.
In section 3.6, “ Therefore, we designed several experiments,” – the sentence seems incomplete (designed experiments to/for?).


**Q5 Detailed Comments To The Authors:**

Questions:

1. In section 3.5, “For MixUpand CutMix, k represents the number of images combined,” how do you combine multiple images (k>2) for Mixup?

2. In section 3.6, “OnSRN18-CIFAR10, we only swept over the top image in StackMix for the first convolutional layer”, could you explain more about the intuition for this step?

3. “In another case, the standard one-hot setup is given two forward passes for inference at test time.” Why are two-forward passes given, and how?


**Q7 Justification For Your Score:**

The paper proposed a novel and simple data augmentation method, StackMix. The authors performed extensive experiments with the ablation study and provided a fair comparison by controlling the number of parameters, epochs, etc., showing a clear performance gain using StackMix on top of other augmentation techniques (Mixup and CutMix).

**Q9 Complying With Reviewing Instructions:**

1: Yes.

---

### Official Review · Reviewer_8vqb · 2022-04-11

**Q2(1) Originality/Novelty:** 2
**Q2(2) Significance/Impact:** 2
**Q2(3) Correctness/Technical Quality:** 3
**Q2(6) Clarity Of Writing:** 3
**Q6 Overall Score:** 5
**Q8 Confidence In Your Score:** 3

**Q1 Summary And Contributions:**

The paper presents a new method for mixing image inputs and outputs, called "StackMix", for data augmentation: each image in this process is a combination of two images (one stacked on top of the other), and the label associated with this combined image is the mean of the one-hot labels. The proposed method can also be combined with other techniques inspired by Mixup, such as CutMix.

**Q2 Assessment Of The Paper:**

More detailed information regarding each of these aspects is given below:

**Q2(4) Quality Of Experiments (Optional):**

3: Good: The experimental evaluation is adequate, and the results convincingly support the main claims.

**Q2(5) Reproducibility:**

2: Fair: Key resources (e.g., proofs, code, data) are unavailable but key details (e.g., proof sketches, experimental setup) are sufficiently well-described for an expert to confidently reproduce the main results.

**Q3 Main Strengths:**

The proposed method is very simple and shows performance improvements across the board in the experiments when combined with other mix-up based techniques. The results show that it is competitive with CutMix on its own and boosts performance further when combined with it (or Mixup).

The compatibility with data augmentation is also evaluated.

Results indicate an improvement with corrupted data and in the semi-supervised learning case as well.


**Q4 Main Weakness:**

There is no theoretical explanation of why the proposed method helps to improve performance. There is a lack of an attempt to motivate the proposed approach in general.

Inference speed is reduced by applying the new technique.

**Q5 Detailed Comments To The Authors:**

It would be very useful to provide some form of justification for the proposed approach that is not just based on improved accuracy on the test set. I can see that it may be difficult to provide theory justifying the proposed method, but perhaps some intuitive justification could be provided.


**Q7 Justification For Your Score:**

The proposed method and very simple and seems worth giving a try in any practical image classification problem. It is one more trick to add to the bag-of-tricks. However, there is no convincing attempt to explain why the proposed approach works (and no theory).


**Q9 Complying With Reviewing Instructions:**

1: Yes.

---

### Official Review · Reviewer_Z7zb · 2022-04-15

**Q2(1) Originality/Novelty:** 2
**Q2(2) Significance/Impact:** 2
**Q2(3) Correctness/Technical Quality:** 2
**Q2(6) Clarity Of Writing:** 2
**Q6 Overall Score:** 3
**Q8 Confidence In Your Score:** 4

**Q1 Summary And Contributions:**

This paper presents a data augmentation method called stackmix for training the deep neural networks. The proposed stackmix method can be utilized with several existing mix-based augmentation methods and achieves a performance improvement.

**Q2 Assessment Of The Paper:**

More detailed information regarding each of these aspects is given below:

**Q2(4) Quality Of Experiments (Optional):**

1: Poor: The experimental evaluation is flawed or the results fail to adequately support the main claims.

**Q2(5) Reproducibility:**

2: Fair: Key resources (e.g., proofs, code, data) are unavailable but key details (e.g., proof sketches, experimental setup) are sufficiently well-described for an expert to confidently reproduce the main results.

**Q3 Main Strengths:**

+ The proposed method has been evaluated on several benchmarks.
+ The idea is simple and the proposed method achieves a performance improvement on several benchmarks.

**Q4 Main Weakness:**

- About the motivation, the key motivation is unclear. As described in the introduction or method sections, the key motivation are “existing mix-based methods cannot be combined with each other” and “most of existing mix-based methods cannot effectively combine more than 3 images”. It is very confused to me that why we need to combine the existing mix-based methods. It is necessary to give a detailed discussion about the motivations theoretically. The authors try to utilize the experimental results about “Mixup-> CutMix” and “Mixup” to show the necessity of the motivation. To my knowledge, the results are weak because Miixp should perform better than “Mixup-> CutMix”. Mixup->CutMix gives a noise cutmix label for a mixup image input.




**Q5 Detailed Comments To The Authors:**

My major concerns are about the key motivations and the experiments of the paper.

-As for the proposed stackmix method, it has the highest performance when $k$ is equal to 2. The results make me confused about the second motivation. Should we need to combined more than 2 images for data augmentation? Please explain for this phenomenon.

- About the paper writing, a lot of grammatical errors exist in the main manuscript and make the paper hard to follow. For example, page 1 “There is work to reduce the cost of the search” -> “There are some works to reduce the cost of the search”, page 2 “Follow up work…with correspondingly weighted labels”-> “Follow-up works… label the image with correspondingly weighted labels”, page 3 “setting -including” -> “setting, inclduing”

- In section of Related work, the difference between the proposed StackMix and the previous Mix-based augmentation method should be discussed in detail.

- The size of the stacked image input is related to the number of $k$. The authors claim that the proposed method do not change the general network architecture. To my knowledge, the size of the input can influence the structure of the network.

- About the experiments, the authors list all of the experiments settings in Table 2. The authors have conducted the experiments on supervised learning setting for several datasets and networks, but only on Cifar dataset for other settings. Can the authors explain for this?

**Q7 Justification For Your Score:**

The concerns come from the unclear motivations, poor presentation, and the confused experimental results of the paper.


**Q9 Complying With Reviewing Instructions:**

1: Yes.

---

### Decision · Program_Chairs · 2022-05-15

**Decision:**

Accept (Poster)

**Comment:**

Meta Review: AC read the paper, reviews, and responses. AC appreciates the simple and effective StackMix method that surpasses all existing baselines.  Though the average rating is below the acceptance bar, AC still recommends acceptance due to the comprehensive experimental results that may shed light on future research in the community. However, AC suggests that the authors do follow the negative comments, especially from Reviewer Z7zb, to improve the quality of the paper for publication.